# Application of a Novel Film Sealant Technology for Penetrating Corneal Wounds: An Ex-Vivo Study

**Jackie Tan [1,*], Leslie John Ray Foster [1,2] and Stephanie Louise Watson [1]**

[1]  The University of Sydney, Save Sight Institute, Discipline of Ophthalmology, Sydney Medical School, Sydney, NSW 2000, Australia; john.foster1@sydney.edu.au (L.J.R.F.); stephanie.watson@sydney.edu.au (S.L.W.)
[2]  Bio/polymers Research Group, Department of Chemistry, The University of Alabama in Huntsville, Huntsville, AL 35899, USA
[*]  Correspondence: jtan8777@uni.sydney.edu.au



**Featured Application: This novel chitosan film sealant is capable of closing penetrating corneal wounds after routine cataract surgeries, closing traumatic corneal lacerations from battlefield injuries, and replacing sutures in corneal grafting procedures.**

**Abstract:** Aim: To compare the burst pressures of corneal wounds closed with a laser-activated, chitosan-based thin film adhesive against self-seal, sutures and cyanoacrylate. Methods: 2, 4 or 6 mm penetrating corneal wounds were created on 100 freshly enucleated bovine eyes. The wounds were closed using a laser-activated chitosan adhesive (n = 30), self-sealed (control) (n = 30), sutures (n = 20) or cyanoacrylate glue (Histoacryl®) (n = 20). The corneoscleral rim was dissected and mounted onto a custom burst pressure testing chamber. Water was pumped into the chamber at 9ml/hr. The fluid pressure prior to wound leakage was recorded as the 'burst pressure'. Results: The burst pressure for the 2, 4 and 6 mm wounds were 239.2 mmHg (SD = ±102.4), 181.7 mmHg (SD = ±72.8) and 77.4 mmHg (SD = ±37.4) ($p < 0.00001$), respectively, for chitosan adhesive. Burst pressure was 36.4 mmHg (SD = ±14.7), 4.8 mmHg (SD = ±4.9) and 2.7 mmHg (SD = ±1.3) ($p < 0.00001$), respectively, for the self-sealed group. For 4 and 6mm wounds, burst pressures with sutures were 33.0 mmHg (SD = ±19) and 23.5 mmHg (SD = ±17.4) ($p = 0.0087$), respectively. For cyanoacrylate, burst pressures for 2 and 4 mm wounds were 698 mmHg (SD = ±240.3) and 494.3 mmHg (SD = ±324.6) ($p = 0.020087$), respectively. Conclusion: This laser-activated chitosan-based adhesive sealed bovine corneal wounds up to 6 mm in length. Burst pressure was higher for the adhesive than sutured or self-sealed wounds, but lower than for cyanoacrylate.

**Keywords:** corneal laceration; sealant; glue; adhesive; chitosan

## 1. Introduction

Full thickness corneal wounds occur with routine cataract surgeries, corneal graft surgeries, penetrating injuries, or as a consequence of severe ocular surface disease. Such full thickness wounds can gape, allowing egress of aqueous fluid, resulting in hypotony, cataract and/or iris prolapse [1,2]. Simultaneously, ocular surface contaminants can enter the eye's internal sterile environment [2,3]. This "recoil suction" effect had been observed via the ingress of blood stain, fluorescein and Indian Ink into the anterior chamber [3–5]. It was postulated that if a significant pathogen load enters the eye, endophthalmitis may occur [2]. Endophthalmitis, though uncommon with rates of 0.1% to 0.17% [6,7] carries a grave ophthalmic prognosis.

For cataract surgery, clear corneal incisions (CCI) had been shown with Ocular Coherence Tomography (AS-OCT) and Indian Ink ingress had been shown to be not truly self-sealing, even when performed by experienced surgeons [3,5]. Accidental manipulation by patients can exacerbate leaks from these wounds and [8] intraocular pressure (IOP) can spike transiently to 110 mmHg with an eyelid squeeze [9]. Wallen et al. reported a 44 times increased risk of endophthalmitis if the CCI leaked on day one post-cataract surgery versus no leak [6]. However, the technical advantages of CCIs have kept cataract surgeons from abandoning this technique [10]. This has prompted research into adjuncts for hermetic corneal wound closure.

There are a variety of adhesives being used to effect corneal wound closure, with many used off-label due to insufficient data from clinical trials [11]. Cyanoacrylate glues were used to seal corneal wounds but are considered "second-line", due to their cytotoxic breakdown products, easy dislodgement prior to complete healing, abrasive nature requiring an overlying bandage contact lens, and non-biodegradability [12,13]. Fibrin glue had also been applied successfully on the corneal surface but has lower repair strength compared to cyanoacrylates, requires longer preparation time and has inherent risk for blood borne infections [14]. ReSure® sealant is a Polyethylene Glycol (PEG)-based hydrogel sealant recently approved for ophthalmic use by the American Food and Drug Administration (FDA) [15]. ReSure® tolerates moderate IOP, is synthetic, biodegradable and remains on the ocular surface for three to seven days [16].

Chitosan is a biopolymer that can be prepared into thin films and holds promise as a material suitable for corneal wound closure. Foster and co-workers had demonstrated that chitosan's naturally adhesive properties can be enhanced through targeted irradiation with a near-infrared laser in the presence of an appropriate chromophore [17–19]. This was postulated to enhance polyanionic-polycationic interaction and hydrogen bonding between polymers of collagen and chitosan during transient heat expansion and subsequent cooling [18,19]. In in-vitro studies, this chitosan adhesive was found to be biocompatible with fibroblast cell-lines [19]. This chitosan adhesive was also able to anastomose transected tibial nerve in an in-vivo rodent model [20]. To assess corneal wound integrity, the highest tolerable IOP prior to wound leak—"burst pressure"—had been used by various groups to assess other ocular surface adhesives [18,21,22]. In this study, we aimed to determine the burst pressure of full thickness corneal wounds closed with this chitosan adhesive and compared against the burst pressures generated from corneal wounds closed with traditional methods of wound closure—sutures and cyanoacrylate glue.

## 2. Methods and Materials

Chitosan adhesive films were supplied by Repartech Pty Ltd., Australian Technology Park, Eveleigh, Australia. These films were 8 mm in diameter, disc-shaped and translucent green due to the chromophore Indocyanine Green (ICG). These films had an average thickness of 30 nm, with a range of 20–40 nm.

### 2.1. Laser Parameters

A continuous-wave Aluminium gallium arsenide (GaAlAs) diode, 810 nm, collimated an infrared invisible laser with a co-axial low power (<0.1 mW), 633 nm, and a red Helium-Neon (HeNe) pointer was used to activate adhesion. The laser spot diameter was 1 mm. This laser was generated by MDL-III-808-1W (serial# 16101161) coupled with PSU-III-LED laser power supply, both manufactured by Changchun New Industries Optoelectronic Tech Co. Ltd. (888 JinHu Road, High-tech Zone Changchun, China). Prior to use, the laser energy was calibrated to 125 mW using a laser power meter (LP100, Serial# 1128CA16L) after zeroing to background radiation. Class 4 laser eye protection was worn by all personnel during the laser application.

### 2.2. Ex-Vivo Tissue

Freshly enucleated bovine eyeballs were obtained daily from our nearest abattoir (Wilberforce, NSW), with a post-mortem time of 4 to 6 hours. Bovine eyeballs were transported on ice to our laboratory (1 hour). On arrival, corneas were screened for previous or current keratitis, corneal scars and other obvious abnormalities under an operating microscope. Only healthy enucleated bovine eyeballs were used in this study.

### 2.3. Wound Creation

A full-thickness central corneal wound was created by a fresh Swann-Morton size 11 scalpel blade, incising along the longitudinal axis of the oval-shaped bovine cornea. The incision was in a single-plane and perpendicular to the corneal surface. Penetration was confirmed by egress of aqueous fluid. Wound sizes of 2, 4 and 6 mm were created, and the wound size was confirmed using Castroviejo callipers externally and internally under the operating microscope.

### 2.4. Study Groups

Full thickness corneal wounds were treated with either (1) Chitosan adhesive, (2) sutures, (3) Histoacryl® cyanoacrylate glue, or (4) left to self-seal. As cyanoacrylate glue is currently not indicated for penetrating corneal wounds larger than 3 mm, only 2 and 4 mm wounds were glued. Wounds 2 mm or less are typically left to self-seal, hence only 4 and 6 mm wounds were sutured.

### 2.5. Chitosan Adhesive Application and Laser

The chitosan adhesive was applied centrally over the 2 mm (n = 10), 4 mm (n = 10), and 6 mm (n = 10) full-thickness corneal wounds after the corneal surface was dried using a cotton-bud. The laser was then manually applied 1–2 mm above and perpendicular to the adhesive, moving circumferentially from outside to inside at a rate of 1 mm/sec under the operating microscope. Activation of the adhesive was confirmed by slight whitening and mild warping of the film onto the corneal surface (Figure 1). Pincher-grip pressure was applied to the cornea to ensure hermetic application of chitosan-film prior to burst pressure testing. If leakage was detected, the adhesive was peeled off and a new adhesive was re-applied.

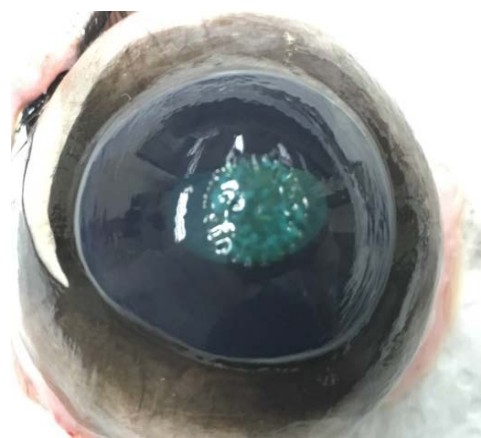

**Figure 1.** An activated chitosan film patch on bovine cornea.

### 2.6. Cyanoacrylate Glue Application

Histoacryl® (B.Braun Surgical S.A Carretera de Terrassa, 121 08191 Rubi, Spain) was drawn up into a 1 mL syringe and delivered to the cornea using a 24-gauge needle to cover the 2 mm (n = 10) and 4 mm (n = 10) wounds under the operating microscope. The cornea surface was thoroughly dried

with a cotton bud prior to glue application. Polymerisation of the glue was instantaneous and difficult to control. As a result, glue application was repeated with a fresh eyeball if the resultant glue did not polymerise uniformly over the wound.

### 2.7. Sutured Wounds

10-0 monofilament non-absorbable nylon sutures with a 5.5 mm 1/2c Spatula needle (Ethilon, Ethicon INC 2007, CS Ultima) were used to close the corneal wounds. Consistent with current clinical practice, penetrating wounds smaller than 2 mm were left to self-seal, two simple interrupted sutures were applied to 4mm wounds (n = 10) and three simple interrupted sutures were applied to 6 mm wounds (n = 10) to ensure a standard space between sutures. Sutures were evenly spaced and applied at 90% corneal depth. All knots were rotated and buried into corneal stroma. All procedures were carried out using an operating microscope.

### 2.8. Control Wounds

Controls were corneal wounds left to self-seal, each n = 10 for 2, 4 and 6 mm groups.

### 2.9. Burst Pressure Testing

The corneoscleral rim was carefully circumferentially dissected at 4 mm posterior to the limbus. The ciliary body, iris and crystalline lens were gently removed from the corneoscleral rim prior to mounting the rim onto a custom-made burst pressure chamber designed for bovine corneas [18]. Water was pumped into the chamber at a rate of 9 mL $h^{-1}$ and fluid pressure inside the mounted rim was digitally measured by a submersible pressure probe (Keller GmbH, Jestetten, Germany). Burst pressure was defined as the highest tolerable fluid pressure recorded prior to wound leak, as evident by a sudden drop in raising fluid pressure and confirmed by fluid egress from the wound.

### 2.10. Statistical Analysis

Data was analysed using an unpaired t-test when comparing means between two independent samples, and a one-way analysis of variance (ANOVA) test when comparing means of three or more independent samples. A *p*-value of 0.05 or less was considered statistically significant. IBM SPSS software version 26 was used to calculate the *p*-values.

## 3. Results

The chitosan adhesive adhered to all corneas tested. It was easy to manipulate the pre-activated adhesive on the ocular surface and the adhesive property could be activated on-demand. Before laser activation, the chitosan adhesive was "sticky" on the dried but moist corneal surface. Once activated, this chitosan adhesive became firmly adherent to the anterior corneal surface.

2, 4 and 6 mm wounds sealed with chitosan adhesive tolerated average burst pressures of 239.2 mmHg (SD = ±102.4), 181.7 mmHg (SD = ±72.8) and 77.4 mmHg (SD = ±37.4), ($p < 0.00001$) respectively (Figure 2). All observed leakages during testing occurred beneath the intact chitosan adhesive. The burst pressures obtained with chitosan adhesive were significantly higher than sutured or self-sealed wounds of identical sizes. However, the burst pressure obtained from cyanoacrylate glue was significantly higher than chitosan adhesive for wounds of identical sizes.

For 2 mm wounds, the average burst pressure from chitosan adhesive was 239.2 mmHg (SD = ±102.4), while self-sealed 2 mm wounds returned an average of 36.4 mmHg (SD = ±14.7). For a wound of the same size closed with Histoacryl® cyanoacrylate glue, the average burst pressure was 698.0 mmHg (SD = ±240.3). There was a statistically significant difference between the three groups ($p < 0.00001$). For 2 mm wounds, chitosan adhesive tolerated higher burst pressure than self-sealed, but lower burst pressure than that attained by cyanoacrylate glue. Figure 3 below illustrates the average burst pressure of 2 mm corneal wounds by each closure modality.

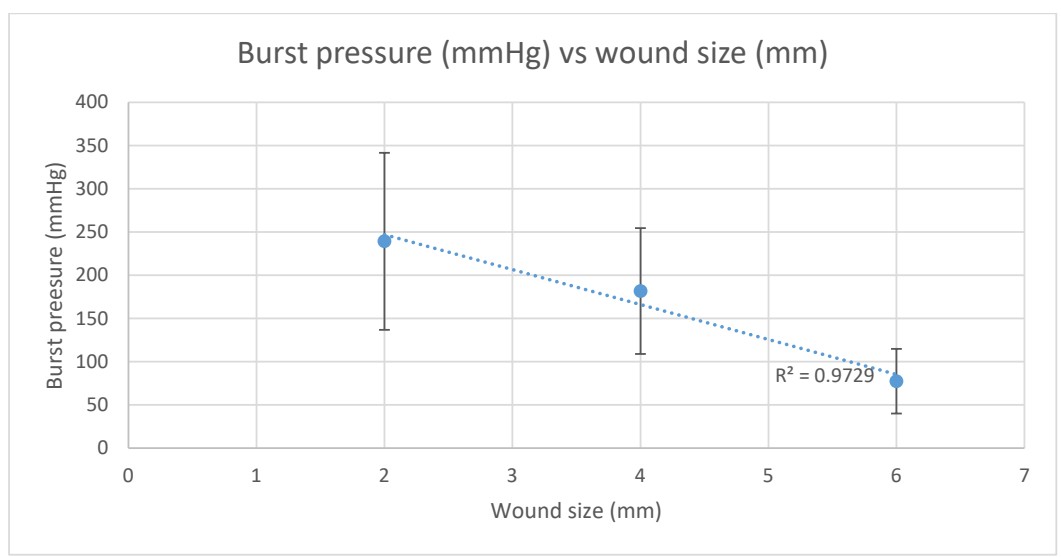

**Figure 2.** Linear relationship of burst pressure (mmHg) with the chitosan adhesive versus wound size (mm). Coefficient of determination ($R^2$) = 0.9729.

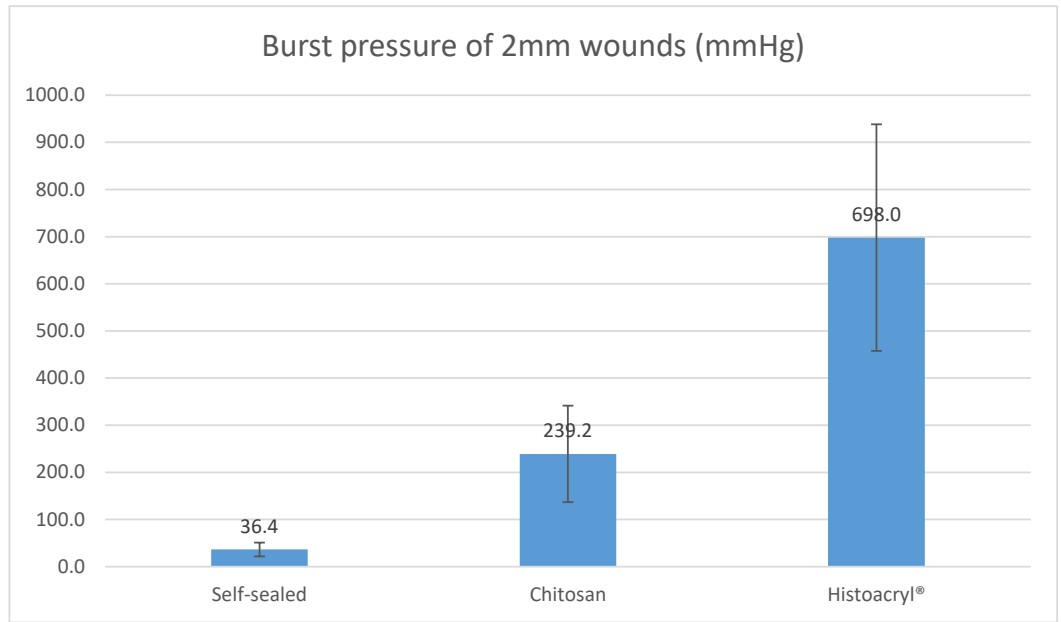

**Figure 3.** The burst pressure of 2 mm corneal wounds closed by self-seal, chitosan and Histoacryl®, $p < 0.00001$.

For 4 mm wounds, the average burst pressure from chitosan adhesive was 181.7 mmHg (SD = ±72.8), while self-sealed wounds returned an average of 4.8 mmHg (SD = ±4.9) and wounds sutured with X2 10-0 nylon generated 33.0 mmHg (SD = ±19.0). For a wound of the same size closed with Histoacryl® cyanoacrylate glue, the average burst pressure was 494.3 mmHg (SD = ±324.6). There was a statistically significant difference between the four groups ($p < 0.00001$). Overall, with 4 mm wounds, chitosan adhesive tolerated higher burst pressure than self-sealed and sutured wounds, but lower burst pressure than that attained by cyanoacrylate glue. Figure 4 below demonstrates the average burst pressure of 4mm corneal wounds by each closure modality.

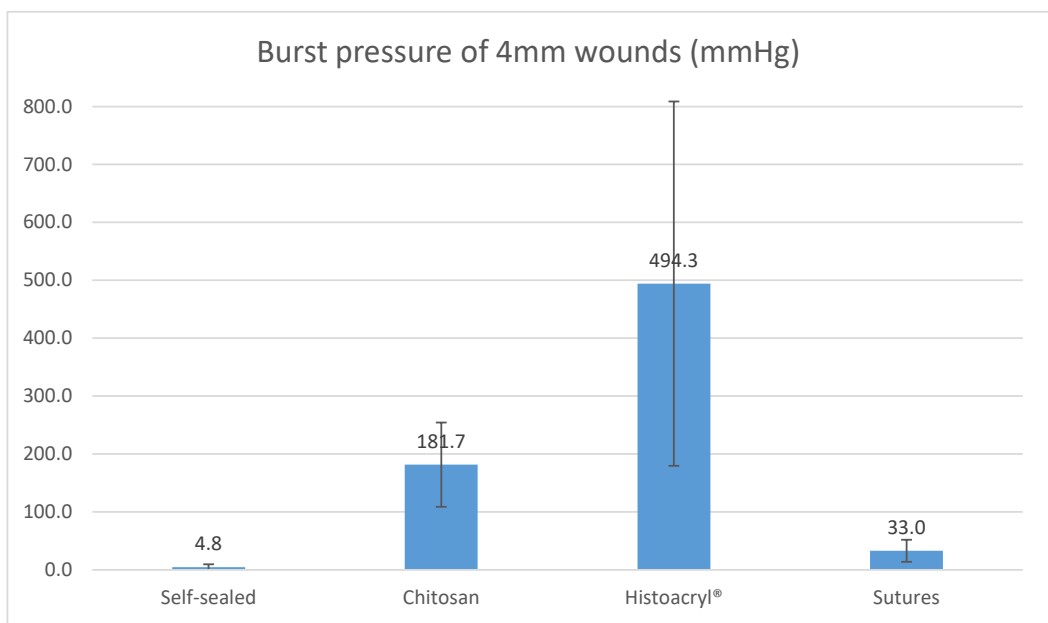

**Figure 4.** The burst pressure of 4 mm corneal wounds closed by self-seal, chitosan, Histoacryl® and sutures, $p < 0.00001$.

For 6 mm wounds, the average burst pressure from chitosan adhesive was 77.4 mmHg (SD = ±37.4), while self-sealed wounds returned an average of 2.7 mmHg (SD = ±1.3). For a wound of the same size sutured with X3 10-0 nylon, the tolerated average burst pressure was 23.5 mmHg (SD = ±17.4). There was a statistically significant difference between the three groups ($p < 0.00001$). For 6 mm wounds, chitosan adhesive tolerated higher burst pressure than self-sealed and sutured groups. Figure 5 below shows the average burst pressure of 6mm corneal wounds by each closure modality.

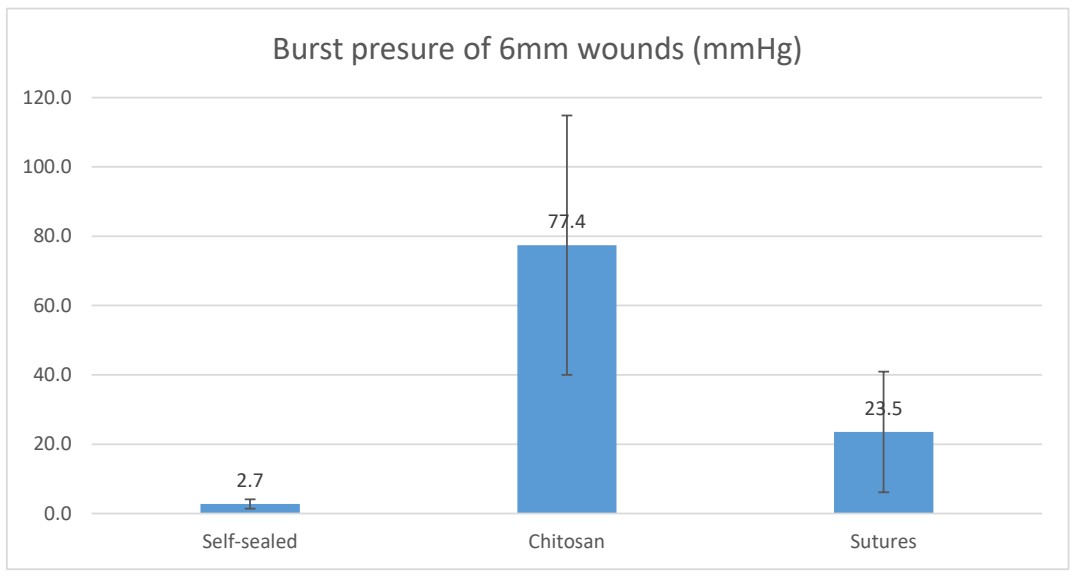

**Figure 5.** The burst pressure of 6mm corneal wounds closed by self-seal, chitosan and sutures, $p < 0.00001$.

The comparative burst pressures between different wounds sizes and treatment modalities is shown in Figure 6. While self-sealed wounds exhibited a relatively modest average burst pressure for 2 mm wounds, they displayed negligible burst pressures for 4 mm and 6 mm sized wounds. The application of sutures was able to increase the burst pressure for 4 mm ($p < 0.00001$) and 6 mm ($p = 0.000246$)

wounds significantly higher than those closed by self-sealing. Treatment with laser-activated chitosan film further increased the burst pressure attained by sutures for 4 mm ($p < 0.00001$) and 6 mm ($p = 0.000033$) corneal wounds. Histoacryl® cyanoacrylate glue produced the highest burst pressure amongst all groups.

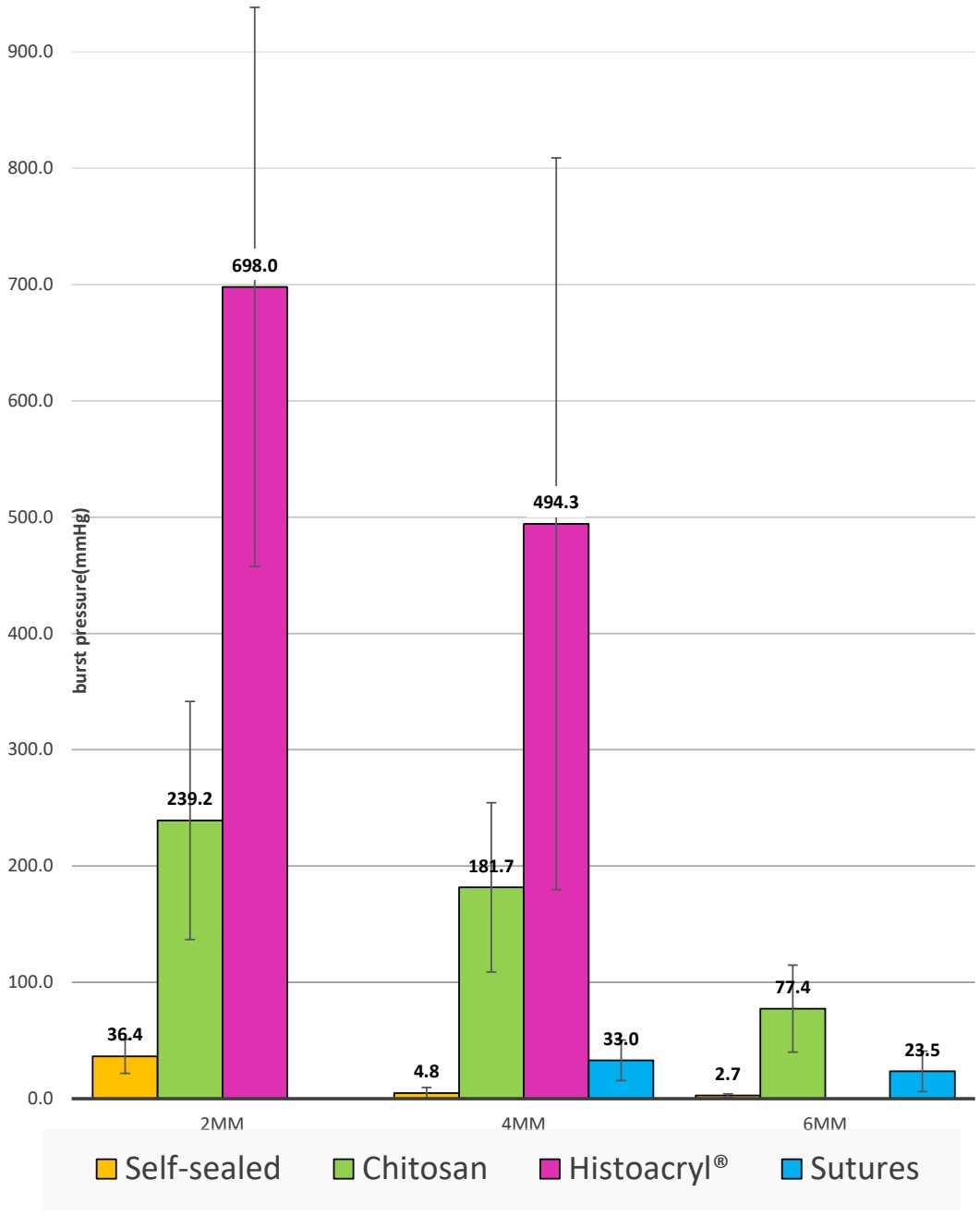

**Figure 6.** The average burst pressures of 2 mm, 4 mm and 6 mm corneal wound closed by self-seal, chitosan adhesive, Histoacryl® and sutures.

## 4. Discussion

This study evaluated a laser-activated chitosan film as a potential surgical device for sealing penetrating corneal wounds and compared the burst pressure against sutures and cyanoacrylate. From our ex-vivo bovine model, we demonstrated that the laser-activated chitosan film was able to adhere to the corneal surface and sealed full thickness corneal wounds. We tested the adhesive against penetrating corneal wounds of 2, 4 and 6 mm in length and recorded average burst pressures ranging from 239.2 mmHg to 77.4 mmHg. The burst pressure of wounds sealed using the chitosan adhesive technology had a linear relationship with wound size—see Figure 2. For each wound size, the chitosan adhesive tolerated significantly higher burst pressures than sutured or self-sealed groups. The chitosan adhesive could not, however, tolerate the higher burst pressures observed in wounds of the same size that were closed with cyanoacrylate glue. However, the cytotoxic nature of cyanoacrylate breakdown products limits its widespread use [12].

An earlier prototype of chitosan adhesive film, SurgiLux®, was also evaluated for its capability to seal full thickness corneal wounds in an ex-vivo bovine model, and tolerated a burst pressure of 251 mmHg (SD = ±10.4) for 2 mm wounds, 328 mmHg (SD = ±12.9) for 3 mm wounds and 224 mmHg (SD = ±16.1) for 6 mm wounds [18]. This result was consistent with the high burst pressure obtained from the chitosan adhesive in our present study. However, Surgilux® was not fabricated in a clean environment and required approximately three times more fluence for activation. Another study evaluating 1 mm trephined corneal perforations used cyanoacrylate with a 3 mm plastic surgical drape patch and found a mean burst pressure of 325.9 mmHg [23] in an ex-vivo human model. This was consistent with the high burst pressures from our Histoacryl® cyanoacrylate groups, which yielded an average of 698.0 mmHg and 494.3 mmHg for 2 and 4 mm wounds, respectively. Similarly, Strassmann et al. applied three 10-0 nylon sutures to 6 mm central corneal wounds and generated a mean burst pressure of 51 mmHg (SD = ±7) [21]. In comparison, the slightly lower burst pressure from our suture group could be attributed to differences in clinicians.

The results here clearly show that the laser-activated chitosan film was capable of adhering to the ocular surface of the bovine eye and sealing full thickness corneal wounds. The adherent property of the chitosan adhesive was postulated to be due to polyanionic-polycationic interaction and hydrogen bonding between polymers of collagen and chitosan, due to transient heat expansion and subsequent cooling [20]. The tensile strength of the chitosan film was previously determined as 31.4 MPa (SD = ±4 MPa) [24]. Previously, in a directly measured IOP study, a voluntary eyelid squeeze raised IOP to 110 mmHg [9]. Our wound that was treated by chitosan adhesive tolerated burst pressures of 239.2, 171.8 and 77.4 mmHg for 2, 4 and 6 mm wounds, respectively, which were significantly higher than our self-sealed control groups, which yielded 36.4, 4.8 and 2.7 mmHg, respectively. From our results, leakage at bursting pressures occurred beneath intact chitosan adhesive, suggesting surface-to-surface adhesion failure as the limitation to further IOP elevation. For 2 and 4 mm wounds, chitosan adhesive tolerated higher pressures than those recorded with eyelid squeezing of 110 mmHg [9].

This study also validated the previous findings by Shahbazi et al. When an earlier prototype of chitosan adhesive was compared against Tisseel® fibrin glue, sutures were reinforced with Tisseel® fibrin glue [18]. Apart from providing clinicians with another alternative for penetrating corneal wound closure, the chitosan film allowed for hermetic closure and tolerated elevated IOP, which were risk factors for post-cataract surgery wound leak [6,8]. Whether this could translate to lower risk of post-operative endophthalmitis will require further studies.

The main limitation of this study stems from its ex-vivo nature and, therefore, its inability to assess healing. Apart from confirming the strong adhesive property on bovine ocular surface, this study did not allow for a dynamic assessment of interaction with other ocular structures, such as the conjunctiva, eyelids and tear-film. The bovine cornea is anatomically different from human cornea, as it has seven or eight histological layers and its central corneal thickness is 1.7 mm to 2.0 mm [25]. The burst pressure results could differ if human cadaver eyes were used. However, due to the ease of handling and

ready availability of bovine eyes, we were able to acquire sufficient data to be statistically meaningful. Our custom-designed burst pressure testing chamber (up to 1000 mmHg recordable) did not limit the burst pressure generated by chitosan film and we reduced human error by using automated pressure recordings.

## 5. Conclusions

The chitosan adhesive was strongly adherent to the bovine corneal surface upon laser activation, sealing full thickness corneal wounds up to 6 mm in size. The average burst pressures were determined to be 239.2, 171.8 and 77.4 mmHg for 2, 4 and 6 mm wounds, respectively. The chitosan adhesive film tolerated significantly higher burst pressures than sutured or self-sealed wounds, but tolerated significantly lower burst pressure than wounds sealed with cyanoacrylate glue.

The possible applications of this chitosan adhesive include closing leaking, penetrating corneal wounds after routine cataract surgeries, closing traumatic corneal laceration from battlefield injuries and replacing sutures during corneal grafting procedures.

**Author Contributions:** Conceptualization, L.J.R.F. and S.L.W.; methodology, J.T., L.J.R.F., and S.L.W.; formal analysis, J.T.; investigation, J.T.; resources, L.J.R.F. and S.L.W.; data curation, J.T.; writing—original draft preparation, J.T.; writing—review and editing, L.J.R.F. and S.L.W.; supervision, L.J.R.F. and S.L.W.; project administration, J.T.; funding acquisition, L.J.R.F. and S.L.W. All authors have read and agreed to the published version of the manuscript.

**Funding:** This project is funded by National Health and Medical Research Council (NHMRC) project grant (APP1067749 to Foster and Watson). Professor Stephanie Watson is a Sydney Medical School Foundation Fellow.

**Conflicts of Interest:** The authors declare no conflict of interest in the preparation of this manuscript.

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
