# Peer review of "Application of a Novel Film Sealant Technology for Penetrating Corneal Wounds: An Ex-Vivo Study"

_applsci, doi:10.3390/app10093193_

Round 1
Reviewer 1 Report
the manuscript suffers from several problems. I suggest to check the statistics, also in the captions of the figures, and to improve the type of statistical tests used. In addition, the quality of the images could be improved.
Finally, in the conclusions section, I recommend making a few more considerations on the possibile applications of the ms observations.
Reviewer 2 Report
The manuscript concerns an important topic.I suggest reviewing the style in order to correct some mistakes.
I also suggest improving the quality of the figures,
as well as the description in the captions
of the utilized statistical tests.
In addition, the authors could add further details
on the possibile applications of their observations.
Round 2
Reviewer 1 Report
I suggest to accept the ms after author's modifications
Reviewer 2 Report
The ms could be accepted